# Treatment Reducing Endothelial Activation Protects against Experimental Cerebral Malaria

**DOI:** 10.3390/pathogens11060643

**Published:** 2022-06-02

**Authors:** Sabrina Mota, Johanna Bensalel, Do Hee Park, Sandra Gonzalez, Ana Rodriguez, Julio Gallego-Delgado

**Affiliations:** 1Department of Microbiology, New York University School of Medicine, New York, NY 10016, USA; smota@umich.edu (S.M.); doheepark@google.com (D.H.P.); sandra.gonzalez@nyulangone.org (S.G.); 2Department of Biological Sciences, Lehman College, The City University of New York, Bronx, New York, NY 10468, USA; jbensalel@gradcenter.cuny.edu; 3Ph.D. Program in Biology, The Graduate Center, The City University of New York, New York, NY 10016, USA

**Keywords:** cerebral malaria, endothelial activation, biomarkers, adjunctive therapy, statins, ARBs

## Abstract

Cerebral malaria (CM) is the most severe neurological complication of malaria caused by *Plasmodium falciparum* infection. The available antimalarial drugs are effective at clearing the parasite, but the mortality rate remains as high as 20% of CM cases. At the vascular level, CM is characterized by endothelial activation and dysfunction. Several biomarkers of endothelial activation have been associated with CM severity and mortality, making the brain vascular endothelium a potential target for adjunctive therapies. Statins and Angiotensin II Receptor Blockers (ARBs) are drugs used to treat hypercholesterolemia and hypertension, respectively, that have shown endothelial protective activity in other diseases. Here, we used a combination of a statin (atorvastatin) and an ARB (irbesartan) as adjunctive therapy to conventional antimalarial drugs in a mouse experimental model of CM. We observed that administration of atorvastatin–irbesartan combination decreased the levels of biomarkers of endothelial activation, such as the von Willebrand factor and angiopoietin-1. After mice developed neurological signs of CM, treatment with the combination plus conventional antimalarial drugs increased survival rates of animals 3–4 times compared to treatment with antimalarial drugs alone, with animals presenting lower numbers and smaller hemorrhages in the brain. Taken together, our results support the hypothesis that inhibiting endothelial activation would greatly reduce the CM-associated pathology and mortality.

## 1. Introduction

In the last two decades, progress towards malaria control through coordinated efforts from international agencies, local governments and private foundations has led to a more than 40% reduction in global malaria incidence. However, these improvements have stagnated in the past few years, with malaria still killing more than 600,000 people every year, two-thirds of whom are children under the age of 5 [1].

Children in sub-Saharan Africa are especially vulnerable to cerebral malaria (CM), the most severe neurological complication of malaria caused by infection with *Plasmodium falciparum*. Mortality caused by CM ranges from 15 to 20% and leaves up to 25% of the survivors with long-term neurological sequelae [2,3]. While available antimalarial drugs are effective at clearing parasites from the blood, they do not have specific effects against CM. Clinicians agree on the need for adjunct therapy specific for CM to rescue patients from death and/or the cognitive impairments of malaria while antiplasmodial therapy kills circulating parasites. To date, CM therapy clinical trials have not demonstrated any favorable results [4].

At the vascular level, brains from patients with fatal CM are characterized by sequestration of parasitized red blood cells in the microvasculature that is associated with the breakdown of the blood–brain barrier [5]. There is also a large body of evidence demonstrating endothelial activation and dysfunction during CM [6]. Importantly, several biomarkers of endothelial activation have been associated with malaria severity and mortality, making endothelial cells a potential target for adjunctive therapies during CM [7]. These biomarkers include angiopoietin-1 (Angpt-1), angiopoietin-2 (Angpt-2), von Willebrand factor, Tie-2, osteoprotegerin (OPG), thrombomodulin-2 and soluble cell-surface adhesion molecules (sCAMs). Among these, Angpt-1 and 2 have been most widely investigated in the context of infectious disease, and there is much evidence for their implication in endothelial dysfunction. Angpt-1 and Angpt-2 are antagonistic ligands of the Tie2 tyrosine kinase receptor. In non-pathogenic conditions, Angpt-1 concentrations are higher than those of Angpt-2, promoting endothelial cell survival; however, in the context of cerebral malaria, Angpt-2 is released by endothelial cells, and its concentration increases to surpass that of Angpt-1, which results in increased permeability of the blood–brain barrier. In this context, vaso-protective treatments that prevent blood–brain barrier disruption may be used during CM to “buy time” while classical anti-*Plasmodium* drugs eliminate the parasite. 

Two drug families have been demonstrated to improve endothelial dysfunction, angiotensin II receptor blockers (ARBs) [8,9] and statins [10], which also present partially protective effects in models of CM. Our previous work described how ARBs, a class of drugs used to treat hypertension, protect the integrity of human endothelial monolayers against *P. falciparum*-induced disruption and significantly increased survival in a mouse model of CM [11,12]. Similarly, others have found partially protective effects of atorvastatin, a drug commonly used to treat hypercholesterolemia that prevents *P. falciparum* cytoadherence and endothelial damage in vitro [13] and also reduces mortality, decreases neuroinflammation and prevents cognitive impairment in mice with CM [11,12,14,15,16].

Therapies using statins and/or ARBs have demonstrated beneficial effects in human-population-based and animal studies for several diseases where endothelial integrity is compromised, including Ebola infection, sepsis, pneumonia and influenza [17,18]. It has been proposed that the beneficial effect of these drugs is mediated by the inhibition of endothelial activation that results in stronger blood vessels, preventing edema and hemorrhages [17,18]. However, this hypothesis could not be confirmed since the levels of endothelial activation were not measured in patients or experimental animals in these studies.

Here, we used a combination of a statin (atorvastatin) and an ARB (irbesartan) as adjunctive therapy to conventional antimalarial drugs to treat CM and explore its effect on biomarkers of endothelial activation in experimental models of CM. Both drugs have been used separately in experimental models of CM alone or in combination with classical anti-*Plasmodium* drugs showing a significant reduction in the mortality rate for CM in mice [11,12].

## 2. Results

The effect of a combination treatment of atorvastatin and irbesartan, which individually strengthen vascular integrity by inhibiting endothelial activation [17,18], was tested in an experimental mouse model. As a first approach, groups of *P. berghei* ANKA-infected C57BL/6J mice received the atorvastatin–irbesartan treatment or only the vehicle from the first day of infection. Atorvastatin has a mild anti-*Plasmodium* effect [19] that was observed in the treated mice (Figure 1A). To separate the anti-*Plasmodium* effect from protective effects against CM, all mice were evaluated for signs of CM and sacrificed when they presented a similar percentage of parasitemia ([11.40, 10.09] vs. [11.40, 9.64] CI 95% for the treated and control group, respectively), which occurred between days 6.5 and 8 for treated mice and days 5.5 and 6.5 for controls. The group of mice receiving the atorvastatin–irbesartan treatment presented significantly less neurological signs of CM, as reflected by lower score values (Figure 1B), indicating that the atorvastatin–irbesartan treatment is effective in preventing the development of experimental CM. 

Analysis of plasma from these mice at the time of evaluation and sacrifice showed that two different markers for endothelial activation, the von Willebrand factor and Angpt-2, were significantly reduced in the group that received the atorvastatin–irbesartan treatment; however, no significant differences were observed for other markers, such as Tie2 and Angpt-1 (Figure 2). The ratio between Angpt-2 and Angpt-1 was also significantly decreased in the treated group versus the control group (Appendix A). 

To better represent the clinical situation, where treatment for CM would be provided only after the appearance of neurological symptoms and in combination with a classical antimalarial drug, mice were treated with chloroquine or artesunate as antimalarial drugs, and the atorvastatin–irbesartan combination as adjunctive treatment for CM. Mice infected with *P. berghei* ANKA started to develop severe signs of CM (score ≥ 3) 5.5 days after infection, when they were randomly assigned to one of two treatment groups: 1) receiving an antimalarial drug (chloroquine or artesunate); or 2) receiving an antimalarial drug plus the atorvastatin–irbesartan combination treatment. Those animals receiving the adjunctive therapy in addition to chloroquine or artesunate exhibited a significantly increased survival ratio—3 to 4 times greater—compared with animals who received the antimalarial drug alone (Figure 3A,B). 

The parasitemia levels showed no differences between the two groups during the study, indicating that the strong anti-*Plasmodium* effect of chloroquine or artesunate overcomes the mild effect of atorvastatin and confirms that the atorvastatin–irbesartan combination treatment does not increase survival through an indirect effect on parasite growth (Figure 3C,D). 

The plasma level of L-lactate was also analyzed in the artesunate-treated mice experiment, since it has been proposed as a prognostic marker of mortality in human patients [20]. Mice that did not survive experimental CM exhibited a significant increase in L-lactate levels when compared with mice that recovered from CM (Figure 4A,C), similar to previous findings in humans [21]. The average L-lactate level in the group of mice treated with adjunctive combination therapy was lower than that of the controls; however, the difference was not statistically significant (Figure 4B,D).

To study the effect of the atorvastatin–irbesartan combination treatment on endothelial cells in the treated mice, the levels of Angpt-2 and Tie-2 were analyzed. The small volume of plasma that could be obtained from live mice (<10 µL/mouse) did not allow for testing additional markers. Both markers were significantly lower in animals treated with artesunate plus the adjunctive combination therapy compared to their counterparts treated with artesunate alone (Figure 5A,B), indicating that the levels of endothelial activation were lower in the groups that received the adjunctive treatment. Not surprisingly, mice that did not survive experimental CM exhibited a significant increase in Angpt-2 and Tie-2 levels when compared with mice that recovered from CM (Appendix A).

Whole-brain sections from all mice were evaluated for presence and size of hemorrhages (Figure 6A). Animals that received the adjunctive combination therapy showed lower density and smaller hemorrhages than those receiving the antimalarial drug alone (Figure 6B–E), which is consistent with the hypothesis that the adjunctive treatment strengthens endothelial integrity. As expected, the mice that succumbed to CM presented the highest density and largest size of hemorrhages in both treatment groups.

## 3. Discussion

The use of ARBs and statins to inhibit endothelial activation in infectious diseases has been proposed as an effective approach to prevent pathological complications derived from endothelial dysfunction [17,18]. Several studies showed significant efficacy of these treatments in improving disease outcome in human patients [22,23,24] and animal models of infectious diseases such as sepsis, influenza, pneumonia, cerebral malaria and Ebola [11,12,25,26,27]. The strengthening of the endothelium has been proposed as the most likely cause of the clinical improvements observed after these treatments [17,18]; however, no measurements of biomarkers of endothelial activation were performed in these studies. 

Although it is well-established that ARBs and statins preserve endothelial barrier function, the mechanism of action is still not fully understood. It has been observed that losartan, a commonly used ARB in the clinic, increases barrier function in brain endothelial cells in vitro by preventing the activation of β-catenin [12]. Statins are known to inhibit the mevalonate pathway, which results in the inhibition of Rho prenylation, preventing endothelial barrier disruption [28].

CM pathogenesis includes brain edema and hemorrhages [29,30]. An increase in several biomarkers of endothelial activation is detectable in CM patients [29,31], presumably reflecting endothelial dysfunction during this syndrome. 

A limitation of the experimental model of CM in mice is that it does not reproduce the interactions between infected erythrocytes and endothelial cells that take place during CM in human patients because erythrocytes infected with rodent malaria parasites are not cytoadherent, as *P. falciparum*-infected ones are. Although the triggers for human and experimental malaria may differ, mouse CM is still considered a representative model for the inflammatory environment, endothelial disruption and the brain damage induced in human CM [32]. Markers of endothelial activation, such as the von Willebrand factor and Angpt-2, are upregulated in both human and experimental CM [33,34,35].

Our observation that Angpt-2 was elevated in the plasma of mice with CM mirrors previous findings in human patients [6]. Other markers of endothelial activation, including Tie2, were also elevated in the plasma of mice with CM. Interestingly, the significant increase in Tie2 levels was only observed when mice began antimalarial treatment after development of CM, which may suggest that Tie2 levels are a late biomarker of endothelial activation in experimental CM. This would be supported by the observation that the median Tie2 level in mice sacrificed upon development of CM symptoms was only 1003 ng/mL (Figure 2B), while the median level for mice that succumbed to advanced-stage CM was 1500 ng/mL. We also observed elevated levels of L-lactate in mice that did not survive CM, similarly to human patients that succumb to CM [21]. These novel observations in mice further support the use of this model for the study of the host endothelium during CM.

The decrease in biomarkers of endothelial activation after the administration of atorvastatin–irbesartan combination indicates that the treatment inhibits endothelial activation that is induced during malaria. Taken together with the decrease in the number and size of hemorrhages in the brain after treatment, these observations strongly suggest that the increased survival induced by the atorvastatin–irbesartan treatment is mediated by the inhibition of endothelial activation, preventing damage of the blood–brain barrier during CM.

Host endothelial cells have been proposed as a therapeutic target for CM, since inhibition of endothelial activation and strengthening of inter-endothelial junctions could prevent the development of cerebral pathology while antimalarial drugs eliminate the parasite. An important advantage of this approach is that it would be effective even when administered after the symptoms of CM have already appeared, which is the usual scenario encountered in the clinic. When patients develop CM symptoms, strong inflammatory responses and significant cytoadhesion of infected erythrocytes in the brain are presumably underway. However, if signaling pathways in brain endothelial cells leading to disruption of inter-endothelial junctions could be inhibited, CM-associated pathology would be greatly reduced. Our results in the mouse CM model support this hypothesis, since we observed dramatic increases in survival, even when treatment was administered after severe neurological symptoms had appeared.

Loss of endothelial integrity is the cause of severe pathological complications in a wide range of diseases, from viral hemorrhagic infections to cardiovascular disorders. It is likely that therapies that strengthen endothelial integrity may have an impact on a wide range of diseases that share this complication.

### Limitations of the Study

There are inherent limitations associated with experimental models of severe malaria since rodent-infecting species of *Plasmodium* do not cause cytoadherence of infected erythrocytes to endothelial cells, as *Plasmodium falciparum* does in human CM. Additionally, due to the limited amount of plasma that can be obtained from mice, not all plasma biomarkers of endothelial activation were tested in every set of experiments. Furthermore, there are additional biomarkers of endothelial activation, such as OPG, thrombomodulin-2 and sCAM, that were not included in this study for the same reasons mentioned above. Further studies are warranted to investigate these and other biomarkers to determine whether they are also affected by treatment with ARBs and statins. 

## 4. Materials and Methods

### 4.1. Cerebral Malaria Experimental Models

C57BL/6J mice (4 weeks old) were obtained from Taconic Farms Inc. (Rensselaer, NY, USA). All animals were maintained under barrier conditions and had free access to water and normal laboratory diet.

Mice were infected by i.p. inoculation of 10^6^
*P. berghei* ANKA-infected erythrocytes. All procedures involving animals followed the Guide for the Care and Use of Laboratory Animals published by US National Institutes of Health (NIH Publication No. 85-23, revised 1985) and were performed with the approval of the Animal Care and Use Committee at the University of New York.

Animals were closely monitored to assess signs of CM following the score chart described by Waknine-Grinberg JH et al. [36] with modifications. Briefly, score of CM was based on appearance (Normal = 0; Coat ruffled = 1; Coat staring/panting = 2) and behavior (Normal = 0; Hunched = 1; Partial paralysis = 2; Convulsions = 3). Accumulated scores of 3 were considered severe cases and scores 4–5 critical cases of CM. 

Mice were treated on the indicated days with irbesartan 50 mg/Kg/day, oral gavage (Cayman Chemical, Ann Arbor, Michigan, USA), atorvastatin 10 mg/Kg/day, oral gavage (Cayman Chemical, Ann Arbor, Michigan, USA), chloroquine 20 mg/Kg/day, i.p. (Sigma, Atlanta, GA, USA) and/or artesunate 5 mg/Kg/day, i.p. (USP, Rockville, MD, USA). Control mice were treated with same volume and route of administration with vehicle (5% sodium bicarbonate in PBS for i.p. and distilled water for oral gavage administration).

Blood samples of 3 μL were obtained by vein tail puncture at the indicated times after infection or from cardiac puncture at end-point. Blood was collected in 1.5 mL microtubes containing 100 μL of heparin 1000 U/mL (SAGENT Pharmaceuticals, Schaumburg, IL, USA). Surviving mice were sacrificed when the parasitemia was completely clear.

### 4.2. ELISAs for Biomarkers of Endothelial Activation

L-lactate was quantified in 50 μL of a 1:300 dilution of plasma and using EnzyFluo L-lactate Assay Kit from Bioassay Systems (Hayward, CA, USA). Plasma concentrations of Tie-2 and angiopoietin-2 were measured using 50 μL of 1/2 and 1/80 dilutions, respectively, by ELISAs (R&D systems, Minneapolis, MN, USA) following manufacturer’s instructions. Angiopoietin-1 concentration was evaluated in 100 μL of a 1/2 plasma dilution accordingly to manufacturer’s instructions (BOSTER, Pleasanton, CA, USA). Plasma level of von Willebrand factor was measured in 50 μL of a 1/28 dilution using SimpleStep ELISA Kit from Abcam (Waltham, MA, USA). All plasma biomarkers of endothelial activation were measured in duplicates.

### 4.3. Brain Histological Analysis

Paraffin-embedded 4 μm brain sections were stained with H&E (Sigma, Atlanta, GA, USA) and scanned using the Leica Biosystems SCN400 whole-slide scanner (Leica, Wetzlar, Germany). Experimental-CM-induced histopathological alterations in the brain (models 1 and 3) were assessed by counting and measuring the size of petechiae and hemorrhages in brain histological sections using the Slidepath Digital Image Hub (DIH) software version 3.1 (Leica, Wetzlar, Germany).

### 4.4. Statistical Analysis

All variables were tested for homogeneity of variance using Barlett test and for normality distribution using Shapiro–Wilk normality test. Variables that met the normality and homoscedasticity criteria were further analyzed using parametric t-test. Variables that did not meet the above criteria were analyzed using Wilcoxon rank-sum test. Variables that showed outliers were analyzed using robust methods (Yuen’s test for trimmed means or t-test using medians when tied values where present) [37,38]. Comparison of survival curves was determined by log-rank (Mantel–Cox) test. P values < 0.05 were considered significant in all tests.

All statistical analyses were conducted in R [39] using the packages WRS2 [40] and survMisc [41]. Graphs were made using Prism version 8.0.2 (GraphPad, San Diego, CA, USA).

## Figures and Tables

**Figure 1 pathogens-11-00643-f001:**
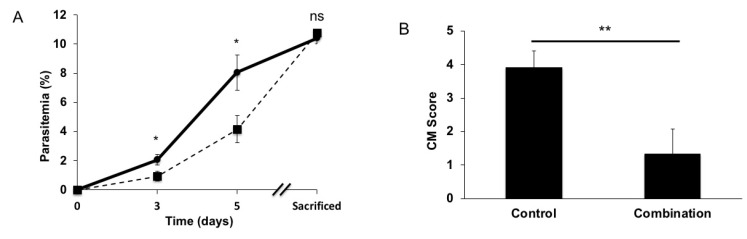
Irbesartan and atorvastatin combination treatment prevented the developing of experimental CM. CB57/B6 mice were treated with irbesartan 50 mg/Kg/day and atorvastatin 10 mg/Kg/day from day one of infection with *P. berghei* ANKA and during the course of the infection (squares, n = 9) versus vehicle-treated control group (circles, n = 10). Animals treated with the combination exhibited lower parasitemia levels and were sacrificed when they presented a similar percentage of parasitemia (days 6.5–8) to the control group (days 5.5–6.5). Student’s t tests were performed, * *p* = 0.0284 and *p* = 0.0166 for days 3 and 5, respectively; results shown as mean ± SEM (**A**). Animals in the combination group (n = 9) showed less neurological signs of CM than control group (n = 10) at the end-point with the same level of parasitemia. Mann–Whitney U test ** *p* = 0.0084; results shown as mean ± SEM (**B**).

**Figure 2 pathogens-11-00643-f002:**
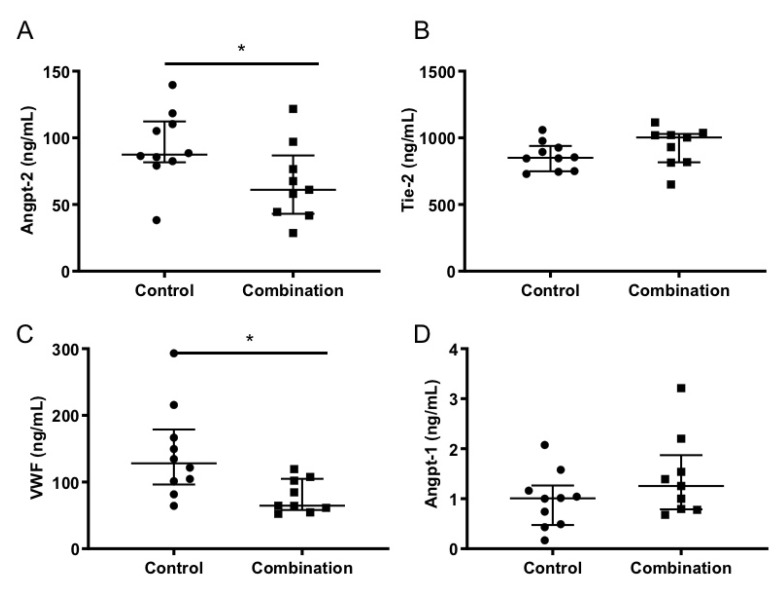
Irbesartan and atorvastatin combination treatment decreased activation of endothelial cells. End-point plasma levels of biomarkers of endothelial cell activation in CB57/B6 mice infected with *P. berghei* ANKA and treated with vehicle (control, n = 10) or with irbesartan 50 mg/Kg/day and atorvastatin 10 mg/Kg/day (combination, n = 9) from day one of infection. Angiopoietin-2 (Angpt-2) (Student’s t test * *p* = 0.0476) (**A**); Tie-1 (**B**); von Willebrand factor (VWF) (Wilcoxon signed-rank test * *p* = 0.0101) (**C**); and Angiopoietin-1 (Angpt-1) (**D**).

**Figure 3 pathogens-11-00643-f003:**
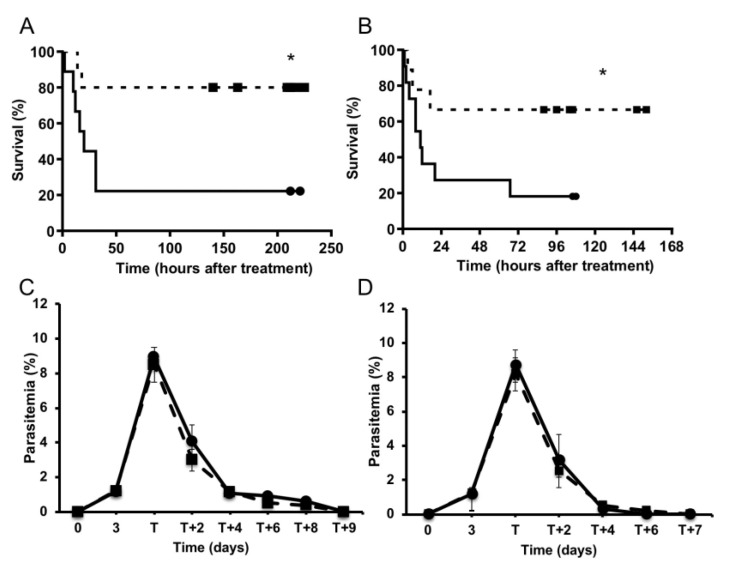
Adjunctive therapy with irbesartan and atorvastatin decreased mortality from cerebral malaria in mice. C57BL/6J mice infected with *P. berghei* ANKA started to develop severe signs of CM (score ≥ 3) 5.5 days after infection, when they were randomly assigned to two treatment groups: 1) receiving an antimalarial drug (circles), chloroquine 20 mg/Kg/day (n = 9) (**A**,**C**) or artesunate 5 mg/Kg/day (n = 9) (**B**,**D**); or 2) receiving an antimalarial drug plus the atorvastatin–irbesartan combination treatment (squares). Those animals receiving the adjunctive therapy in addition to chloroquine (**A**, squares) or artesunate (**B**, squares) had significantly increased survival compared with the antimalarial drug alone (**A**,**B**, circles). Symbols mark the time of mice euthanasia when parasitemia were completely cleared. Parasitemia levels were comparable between **C**) chloroquine-alone group (circles) and the chloroquine-plus-combination-therapy group (squares) or **D**) artesunate-alone group (circles) and artesunate-plus-combination-therapy group (squares). X-axis value “T” is the starting point for all treatments coinciding with the development of severe signs of CM individually monitored for each mouse. Student’s t tests performed, and means ± SEM are shown. Survival curves were determined by log-rank (Mantel–Cox) test. * *p* = 0.0157 and * *p* = 0.0487 for (**A**) and (**B**), respectively, vs. control.

**Figure 4 pathogens-11-00643-f004:**
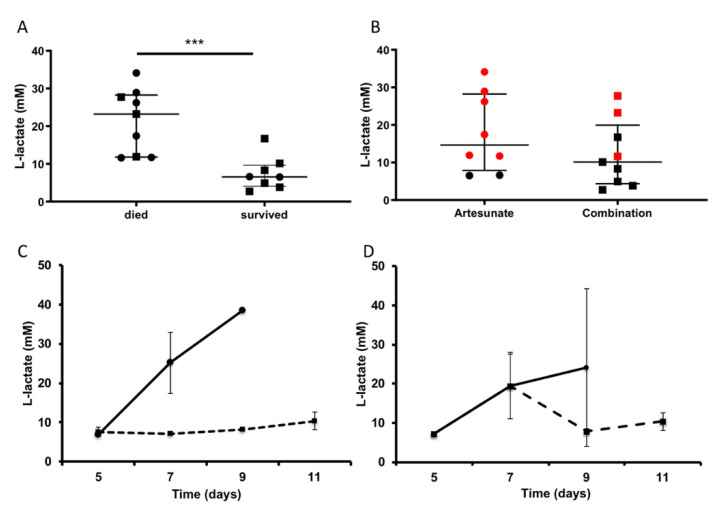
L-lactate plasma levels as a surrogate marker of mortality. Plasma levels of L-lactate at the end-point (**A**,**B**) or throughout infection (**C**,**D**) were determined in C57BL/6J mice infected with *P. berghei* ANKA and treated after developing signs of CM with artesunate 5 mg/Kg/day alone (**A**,**B**, circles, n = 8) or artesunate plus the atorvastatin–irbesartan combination treatment (**A**,**B**, squares, n = 9). Wilcoxon signed-rank test, *** *p* = 0.00058. (**B**) Animals that succumbed to CM (red markers) or survived (black markers) are indicated. (**A**,**C**) Comparison of mice that died (**C**, continuous line) or survived (**C**, dashed line). (**B**,**D**) Comparison of mice treated with artesunate alone (**D**, continuous line) or artesunate plus additional atorvastatin–irbesartan combination (**D**, dashed line). Results are shown as mean ± SEM.

**Figure 5 pathogens-11-00643-f005:**
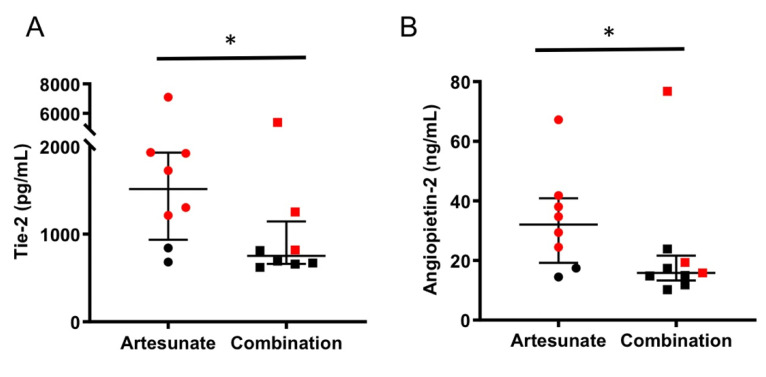
Adjunctive therapy decreased levels of markers of endothelial cell activation in mice with CM. End-point plasma levels of Tie-2 (**A**) and Angiopoietin-2 (**B**) in C57BL/6J mice infected with *P. berghei* ANKA and treated after developing signs of CM (score ≥ 3) with artesunate 5 mg/Kg/day alone (circles, n = 8) or artesunate plus the atorvastatin–irbesartan combination treatment (squares, n = 9). Animals that died from CM or survived are represented with red and black markers, respectively. Yuen’s test, * *p* = 0.03369 and * *p* = 0.03268 for (**A**) and (**B**), respectively.

**Figure 6 pathogens-11-00643-f006:**
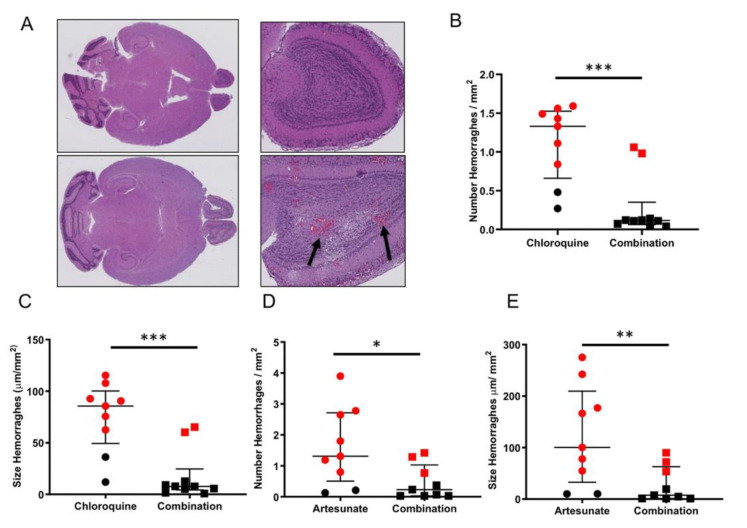
Adjunctive combination therapy decreased the number and size of brain hemorrhages. Histological sections of the brains of mice (**A**) were analyzed for the presence (**B**,**D**) and size (**C**,**D**) of hemorrhages. (**A**) Histological section of a mouse receiving artesunate plus combination treatment (top panel, n = 9) or only-artesunate treatment (lower panels, n = 9). Detail of hemorrhages (arrows) visible at higher magnification in the olfactory bulbs (lower right panel) of one mouse treated only with artesunate. Mice receiving the combination therapy (n = 9) showed lower number and smaller hemorrhages than those receiving chloroquine (**B**,**C**) or artesunate (**D**,**E**) alone. Animals that died from CM or survived are represented with red and black markers, respectively. Wilcoxon signed-rank tests performed, *** *p* = 0.0006, * *p* = 0.0242 and ** *p* = 0.0078.

## Data Availability

Data set will be made available on reasonable request to the study principal investigator (julio.gallegodelgado@lehman.cuny.edu).

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
