# Peer review of "Treatment Reducing Endothelial Activation Protects against Experimental Cerebral Malaria"

_pathogens, 2022, doi:10.3390/pathogens11060643_

Round 1
Reviewer 1 Report
In this manuscript the authors first evaluate endothelial markers (Ang 1 and 2, Tie 2 and vwF) in mice administered the combination of atorvastatin and irbesartan at the same time as being infected with P. berghei ANKA. They report that the endothelial activation markers are lower in the mice who received atorvastatin/irbesartan compared to mice who did not. Second, they evaluate the effect of administering this combination as an adjunct to antimalarial treatment (chloroquine or artesunate) to mice already showing signs of CM. In this 2nd experiment, mice who received atorvastatin and chloroquine had greater survival than mice who received antimalarial treatment alone.
This study adds to existing data from this group and others demonstrating a benefit of ARBs and statins in experimental CM. One difference of this study is that markers of endothelial activation are measured. The paper is well written and provides a useful contribution to the literature.
Comments:
- - The specific endothelial activation markers evaluated (Ang 2, Ang 2, vwf and Tei 2) are not mentioned at all in the background. It would be useful to provide an overview of the Ang-Tie 2 axis, and the different mechanisms/actions of the selected markers (particularly as the actions of each of these markers are quite different), and the reason that these particular markers were selected. The authors mention that markers of endothelial activation have been associated with mortality, but they should mention which markers they are referring to (eg Ang 2 is increased with mortality, but not Ang 1). Also there are other endothelial activation markers that weren’t selected and have been associated with mortality in CM, eg. OPG. Did the authors consider measuring OPG?
- - Could number of mice be mentioned, for each experiment – either in the text, figures, or figure legends. Numbers are given in legend of Fig 3 but not in any other sections.
- - could exact P values be given, rather than just “<0.01” or “<0.05”?
- - The authors report that the lower CM scores may be mediated by lower levels of Ang 2 and wvf. Was CM score correlated with either of these markers of endothelial activation?
- - The authors analysed lactate as a marker of mortality, and found, not surprisingly, that lactate was higher in mice who died compared to mice who survived. Did they also look at the levels of Ang 2 and vWf in mice who survived compared to those who died?
- - Some rewording of the lactate paragraph is required – suggest avoid use of the word “drastic”, and also reword the sentence that refers to “mice that did not survive in this assay”. The next sentence is also slightly clumsy and could be reworded.
- - Suggest restructuring the results section so that the plasma markers are all reported together (ie. Lactate, ang 2, and Tie 2), rather than reporting the lactate first, then the histological findings, then the ang 2 and Tie-2.
- - Given that in the first experiment, vwf, but not Tie2, was significantly reduced in the combination group vs the control group, why did the authors choose to evaluate only ang 2 and Tie2 in the 2nd experiment, but not vwf?
- - What was the rationale for using two different types of antimalarial treatment – chloroquine and artesunate?
- - It would be useful in the discussion for the authors to discuss in more detail the specific findings related to the different markers (ie. Could they speculate on why Tie-2 was reduced in the 2nd experiment, but not in the 1st). Also a limitations section should be added, including the fact that not all markers were measured in both experiments, and also that other markers of endothelial activation (eg. OPG) were not measured.
Reviewer 2 Report
In this manuscript, Mota et al test a combined therapy of antimalarials with antimalarials and statins and angiotensin II receptor blockers in an experimental malaria model. The manuscript is timely and is a natural progression of the previous work of the corresponding authors. Overall the experiments described are logical and sound, and the manuscript is easy to read and well written. However, I believe that some clarifications, especialy in the figures, are needed:
Major
- The authors always applied combinations of antimalarials and atorvastatin+ irbesartan. Did the authors test atorvastatin and irbesartan independently in combination with antimalarials? Why did the authors decided to combine the statin and the angiotensin II receptor blockers?
- Figure 2: Did the authors check for differences in Ang-2 and Ang-1 ratio, which is often dysregulated in clinical cerebral malaria in humans.
- Figure 3: The authors show a poor survival (only 20%) of infected mice after administration of chloroquine and artesunate alone. Is this common in ECM models despite de decrease in parasitemia? In Figure 3B the authors indicate that “symbols mark the time of mice euthanasia when parasitemia was/were cleared” However, graph 3B shows that all artesunate treated animals were euthanized at day 4 and 3D shows parasitemia data until day 9?
- The authors do not indicate what type of deviation they show in the figures. Since many statistical tests were performed as shown in the methods section, the authors should indicate which test was used in each figure legend.
- Why did chloroquine treatment was included in certain figures (3 and 5) and not in others (4 and 6)?
- Discussion: “CM pathogenesis includes brain endothelial activation and dysfunction, which results in brain edema and hemorrhages [28]” The causality between endothelial activation and edema has not been proven. Please rephrase.
- Discussion: “The experimental model of CM in mice does not reproduce the interactions between infected erythrocytes and endothelial cells that take place during CM in human patients because erythrocytes infected with rodent malaria parasites are not cytoadherent, as P. falciparum-infected ones are.” The authors should make it clear that this is a limitation, now it seems vague.
- Discussion “However, mouse CM is considered a representative model for the inflammatory environment, endothelial disruption and the brain damage induced in human CM [30]. Markers of endothelial activation, such as von Willebrandt factor and angiopoietin-1, are similarly modulated in both human and experimental CM [31-33].” The sentence is inaccurate and might give the wrong impression to the unexperienced reader. Although it is true that endothelial activation is shown in both, the molecular mechanisms that trigger endothelial activation/inflammation on human and experimental malaria might have different origins, and therefore might have different “modulation”. Note: there is a typo on von Willebrand.
Minor:
The authors might want to show the legend next to the graphs. The figure legends are long and this will make them understandable at a glance.
